# Assessment of Rapid MinION Nanopore DNA Virus Meta-Genomics Using Calves Experimentally Infected with Bovine Herpes Virus-1

**DOI:** 10.3390/v14091859

**Published:** 2022-08-24

**Authors:** Gaelle Esnault, Bernadette Earley, Paul Cormican, Sinead M. Waters, Ken Lemon, S. Louise Cosby, Paula Lagan, Thomas Barry, Kate Reddington, Matthew S. McCabe

**Affiliations:** 1Animal and Bioscience Research Department, Animal & Grassland Research and Innovation Centre, Teagasc, Oak Park, R93 XE12 Carlow, Ireland; 2Nucleic Acid Diagnostics Research Laboratory (NADRL), Microbiology, School of Natural Sciences, National University of Ireland, H91 TK33 Galway, Ireland; 3Veterinary Sciences Division, Agri-Food and Biosciences Institute, Stormont, Stoney Road, Belfast BT4 3SD, UK; 4Microbial Diagnostics Research Laboratory (MDRL), Microbiology, School of Natural Sciences, National University of Ireland, H91 TK33 Galway, Ireland

**Keywords:** Oxford Nanopore Technologies, MinION, Epi2ME, rapid viral metagenomics diagnostics, bovine herpesvirus 1, bovine respiratory disease

## Abstract

Bovine respiratory disease (BRD), which is the leading cause of morbidity and mortality in cattle, is caused by numerous known and unknown viruses and is responsible for the widespread use of broad-spectrum antibiotics despite the use of polymicrobial BRD vaccines. Viral metagenomics sequencing on the portable, inexpensive Oxford Nanopore Technologies MinION sequencer and sequence analysis with its associated user-friendly point-and-click Epi2ME cloud-based pathogen identification software has the potential for point-of-care/same-day/sample-to-result metagenomic sequence diagnostics of known and unknown BRD pathogens to inform a rapid response and vaccine design. We assessed this potential using in vitro viral cell cultures and nasal swabs taken from calves that were experimentally challenged with a single known BRD-associated DNA virus, namely, bovine herpes virus 1. Extensive optimisation of the standard Oxford Nanopore library preparation protocols, particularly a reduction in the PCR bias of library amplification, was required before BoHV-1 could be identified as the main virus in the in vitro cell cultures and nasal swab samples within approximately 7 h from sample to result. In addition, we observed incorrect assignment of the bovine sequence to bacterial and viral taxa due to the presence of poor-quality bacterial and viral genome assemblies in the RefSeq database used by the EpiME Fastq WIMP pathogen identification software.

## 1. Introduction

Bovine respiratory disease (BRD) is the leading cause of morbidity, mortality and economic loss in cattle of all ages [1,2,3,4,5]. The extensive use of vaccines against BRD-associated viral and bacterial pathogens has not reduced the incidence or severity of BRD in cattle. Consequently, large quantities of broad-spectrum antimicrobials are used for the therapeutic treatment of BRD in Europe and the USA [6,7]. Easy and economical methodologies that enable the rapid and reliable on-farm detection of viral and bacterial pathogens are required to inform rapid targeted treatment and up-to-date vaccine design.

BRD is a multifactorial disease that is associated with an ever-increasing number of species and strains of viruses and bacteria. Viruses that are commonly associated with BRD cases include bovine herpesvirus type 1 (BoHV-1), bovine parainfluenza 3 virus (BPI-3), bovine respiratory syncytial virus (BRSV), bovine viral diarrhoea virus (BVDV), bovine coronavirus (BCoV) and bovine adenovirus (BAdV) [8,9]. BRD-associated bacteria commonly include *Pasteurella multocida*, *Mannheimia haemolytica*, *Histophilus somni* and *Mycoplasma* spp. [8,9,10]. 

New pathogens, e.g., influenza D virus (IDV) [11] and *Sneathia amnii* [12], are continually added to the list of BRD aetiologies. Viruses are known to initiate the disease by weakening the animal’s defenses, which leads to secondary bacterial infection [13]. Viruses with RNA and DNA genomes have been associated with BRD [8]. 

A diagnosis of BRD is generally based on clinical signs that are assessed via visual inspection of the animal, measurement of rectal temperature and pulmonary auscultation [2,3,14,15]. If the identification of the causative pathogen(s) is attempted (which is often not the case), a nasal swab from the affected animal is sent to a diagnostic laboratory, where a targeted qPCR diagnostic analysis of only four or five of the most likely (e.g., previously mentioned) bacterial or viral pathogens is conducted [16,17]. As there are at least 40 possible bacterial and viral pathogens associated with BRD, it is too expensive and time-consuming to test for all of these using targeted real-time quantitative PCR (qPCR) diagnostics. Consequently, there are considerable delays in receiving the results of the aetiological diagnoses of BRD cases, which are often inconclusive [18]. For pathogen identification to be of direct practical use in preventing a BRD outbreak, the results would need to be available to a veterinarian within 24 h. As such, qPCR falls far short of what is required for rapid BRD-associated pathogen identification. This has prompted the increased application of viral metagenomic approaches that are based on next-generation sequencing (NGS) (e.g., Illumina platform) [8,9] and third-generation sequencing (TGS) (e.g., Oxford Nanopore Technologies platform) [19] to BRD-associated virus diagnostics. However, the relatively high cost and lack of portability of even the smallest Illumina platform NGS machine (the iSeq 100) restrict its availability and use to mainly large, well-funded laboratories. In addition, the sequencing runs on Illumina NGS machines usually take longer than 24 h and sequences can only be viewed and analysed after the run is complete.

In contrast, the Oxford Nanopore Technologies (ONT) MinION DNA/RNA sequencer is a portable, low-cost sequencing device that allows real-time data analyses [20] and has the potential to enable untargeted same-day (sample-to-result) viral diagnostics (i.e., identify viruses to species/strain/sequence-variant level and generate a viral sequence for the assembly of new viral genomes) in small static or mobile veterinary surgeries. Sequencing all of the nucleic acid in a sample potentially allows for the detection, in a single assay, of all organisms, including pathogens, that are present in that sample [20]. On the ONT MinION, DNA or RNA is sequenced on a disposable flowcell that contains a membrane that is covered in active biological nanopores. There are three types of flowcell that can currently be used on a MinION. These are called (1) FLO-MIN106D (R9.4.1), (2) FLO-FLG001 (R9.4.1) and (3) FLO-MIN112 (R10.4). R9.4.1 and R10.4 refer to the types of biological pores in the membrane in each type of flowcell (https://store.nanoporetech.com/eu/flow-cells.html (accessed on 21 April 2020)). The membrane is flanked on either side by opposing electrical charges, which drive negatively charged individual single-chain strands of DNA or RNA through the nanopores towards the positive charge. A sensor registers the unique change in the current produced by bases as they pass through the nanopore. These changes in current are translated into nucleotide sequence information, in the form of FASTQ files, by neural network basecallers [21,22]. As soon as these FASTQ files are generated, they can be uploaded to a cloud-based software platform called Epi2ME, which contains several intuitive point-and-click sequence analysis applications called ‘workflows’.

The objective of the present study was to assess whether the ONT MinION device, Epi2ME cloud-based software and library preparation kits could achieve sufficient sensitivity, accuracy, specificity and speed to correctly detect and generate the genomic sequence of a known BRD virus in nasal swabs, which are the most commonly collected sample in BRD outbreaks, from infected cattle within 24 h. For this, we first tested and optimised ONT procedures using bovine foetal lung cell cultures (bFLC) that were infected with the BRD-associated virus bovine herpes virus-1 (BoHV-1), as, unlike nasal swabs from BoHV-1-infected cattle, the supply of cell cultures was not a limiting factor. BoHV-1 is an enveloped DNA virus with a large genome of 135.3 kb with a high GC content of 72% and is of major economic importance to the cattle industry [23]. We then applied these procedures to nasal swabs that were collected from Holstein–Friesian calves that had been experimentally challenged with BoHV-1.

## 2. Materials and Methods

### 2.1. Bovine Foetal Lung Cells Infected with BoHV-1

Bovine foetal lung cells (bFLCs) were isolated from a bovine foetus. The source, origin and characteristics of these bFLCs are shown in Appendix A. BoHV-1 strain 2011-415 is a virulent field sample that was isolated from a typical fatal clinical case (9-month-old calf) of infectious bovine rhinotracheitis with pulmonary complications. A post-mortem trachea sample from this animal tested positive via qPCR for BoHV-1 and the virus was isolated and stored at −80 °C. A T75 flask (Thermo Fisher Scientific, Agawam, MA, USA) containing bFLC was infected with the BoHV-1 strain 2011-415 isolate with a multiplicity of infection (MOI) of 1. As a negative control, an additional T75 flask containing bFLCs and 5 mL of 2% buffer G-MEM was also used. The two flasks were incubated at 37 °C for 90 min in a CO_2_ incubator (Thermo Fisher Scientific, Carlsbad, CA, USA); then, 15 mL of 2% G-MEM buffer was added. The flasks were replaced in the CO_2_ incubator at 37 °C. At 48 h post-infection, the two flasks were observed via phase-contrast light microscopy at 40× magnification, with 90% of the cells displaying viral cytopathology. The flasks were then placed in a freezer at −80 °C for 2 h, and subsequently thawed and their contents transferred to sterile 50 mL centrifuge tubes (Thermo Fisher Scientific, Agawam, MA, USA). The tubes were then centrifuged for 5 min at 3660× *g*, and 1 mL aliquots of the supernatant were transferred to sterile 1.5 mL microfuge tubes (Eppendorf, Hamburg, Germany) and frozen at −80 °C.

### 2.2. Experimental Calves

All animal experiments were carried out in accordance with the UK Animals (Scientific Procedures) Act 1986 and with the approval of the Agri-Food and Biosciences Institute Northern Ireland Ethical Review Committee. The study was reported in accordance with ARRIVE guidelines [24]. 

As part of a larger study, 12 Holstein–Friesian bull male calves (mean age 21.3 weeks, s.d. ± 3.4) were selected from a larger group of 43 Holstein–Friesian bull calves. The enrolment of calves for the challenge study was based on two criteria: (i) a low level of BoHV-1 antibody and (ii) a negative BoHV-1 qPCR result for a nasal swab collected two weeks before the challenge. The 12 selected calves were assigned to two groups (BoHV-1-challenge and PBS-challenge), with 6 calves per group. For the PBS-challenge group (n = 6), the mean age = 21.4 (s.d. ± 3.3) weeks, mean weight = 173.3 (s.d. ± 23.7) kg and mean BoHV-1 antibody = 18.0 (s.d. ± 4.5%). For the BoHV-1-challenge group (n = 6), the mean age = 21.0 (s.d. ± 4.5) weeks, mean weight = 175.8 (s.d. ± 35.6) kg and BoHV-1 antibody = 20.6 (s.d. ± 13.2%). On day 0, each calf in the BoHV-1-challenge group was experimentally challenged via intranasal atomisation with a 1.35 mL solution of BoHV-1culture. The animals in the PBS-challenge group were mock-challenged (day 0) with intranasal atomisation of 1.35 mL of a sterile PBS solution. The two groups were housed in two separate biocontainment level 3 sheds, each with a 10 m × 5 m floor covered in straw. Daily clinical assessments, nasal swabs, and blood samples were collected from each animal on day −1, day 0, day 1, day 2, day 3, day 4, day 5 and day 6 relative to the day of the challenge. Animals were euthanised on day 6 post-challenge. 

### 2.3. Nasal Swabs from Experimental Calves

For each nasal swab sample, the exterior of the nasal nares of the calf was sterilised with 70% ethanol; then, a sterile swab was removed from its sterile tube and inserted approximately 20 cm into the nostril and rolled on the internal nasal membrane for approximately 5 s. The swab tip was cut with scissors (sterilised with 70% ethanol) into a 2 mL sterile tube and immediately frozen on dry ice. This was repeated once for each animal so that two nasal swabs (nasal swab-1 and nasal swab-2) were taken for each animal on every day of the 8 day trial. Nasal swabs from the BoHV-1 calf challenge model were stored for approximately 6 months at −80 °C prior to nucleic acid extraction qPCR and sequencing. 

Immediately prior to nucleic acid extraction, nasal swabs in tubes were removed from storage in a −80 °C freezer to a class 2 biological safety cabinet. A volume of 1.5 mL of molecular grade PBS (Sigma Aldrich, St. Louis, MO, USA) was added to each tube and the tubes were vortexed for 1 min to release the nasal material from the swabs. The resulting nasal swab eluate was then transferred to a sterile 15 mL tube and the swab remained in the 2 mL tube. A further 1.5 mL of PBS was added to each of the swabs in the 2 mL tubes, the tubes were vortexed again, and this second PBS nasal swab eluate was removed and added to the first 1.5 mL PBS nasal swab eluate in the 15 mL tube, resulting in 3 mL of nasal swab eluate for each swab sample.

### 2.4. Non-Viral Nucleic Acid Depletion

For the bead beating step, either 250 µL of BoHV-1 infected bFLC in vitro culture or 1 mL of nasal swab-2 eluate was transferred to a Pathogen Lysis Tubes L (Qiagen, Manchester, UK). For the negative extraction control, 250 µL of molecular grade phosphate-buffered saline (PBS, pH 7.4) (Sigma Aldrich, St. Louis, MO, USA) was added to a Pathogen Lysis Tube L (Qiagen, Manchester, UK). To prevent the escape of aerosols from the tubes during bead beating, tube lids were sealed with Parafilm (Sigma Aldrich, St. Louis, MO, USA). Tubes were placed in a FastPrep-24 disruptor instrument (MP Biomedicals, Irvine, CA, USA) and shaken at high speed (4 ms^−1^) for 30 s. The tubes were then removed and centrifuged at 500× *g* for 45 s to collect the contents at the bottom of the tube. The supernatant from each Pathogen Lysis Tube was then carefully transferred to 2 mL DNA LoBind Safelock tubes (Eppendorf AG, Hamburg, Germany) and the volume was adjusted to 1 mL with molecular grade PBS. An aliquot of 2.5 µL of RNaseA (4 mg/mL) (Promega, Madison, WI, USA) was added to the tubes, which were then incubated for 15 min at 37 °C in an Eppendorf Thermostat Plus (Eppendorf AG, Hamburg, Germany). Turbo DNase (10 μL) and 10× Turbo DNase buffer (100 μL) were then added (Thermo Fisher Scientific, Carlsbad, CA, USA) [25] and the tubes were gently mixed by pipetting six times and incubated for 30 min at 37 °C in an Eppendorf Thermostat Plus. A further 10 μL of Turbo DNase was added and the contents were again mixed by gently pipetting six times and incubated for a further 30 min at 37 °C. DNase inactivation reagent (112.5 μL) (Thermo Fisher Scientific, Carlsbad, CA, USA) was then added to the samples and mixed by gentle pipetting six times, and incubated for 5 min at 24 °C. The mixture was centrifuged at 10,000× *g* for 90 s; then, the supernatant containing the nucleic acid was transferred to a fresh tube without disturbing the pellet of DNase Inactivation Reagent. Samples were also processed with the omission of certain treatments to test whether these treatments were necessary. Five such treatment regimes were tested. These were: (A) no bead beating or nuclease treatment; (B) bead beating only; (C) no bead beating, 1× RNase and 1× DNase; (D) no bead beating, 1× RNase and 2× DNase; and (E) bead beating, 1× RNase and 2× DNase. A negative extraction control (a tube containing only molecular grade PBS, or a sterile unused swab) was included with each batch of extractions to monitor the contamination of reagents and cross contamination of samples during the extraction process. 

### 2.5. Nucleic Acid Extraction and Purification

For nasal swab-1 eluates, nucleic acid was extracted with a Roche Magnapure using a Roche Total Nucleic Acid Isolation Kit (Roche, Basel, Switzerland). Nucleic acid extraction and purification from nasal swab-2 eluates and bFLCs were performed using the QIAamp UltraSens Virus Kit (Qiagen, Manchester, UK) according to the manufacturer’s instructions [26] with the exception of the substitution of 5.6 µL of carrier RNA with 5.6 µL of a solution of 5 mg/mL linear acrylamide. Total nucleic acid was immediately extracted and purified from the swab eluates and bFLCs following the nuclease treatment.

### 2.6. qPCR Analysis

qPCR analysis was performed in triplicate and the mean of the three resulting Cq values was used if the standard deviation was within the default set limits of ABI7500 software (Applied Biosystems, Waltham, MA, USA). The BoHV-1 *UL27* gene TaqMan^®^ Custom Gene Expression Assay (Applied Biosystems, Waltham, MA, USA) comprised BoHV-1 F (forward primer) 5′-TGT GGA CCT AAA CCT CAC GGT-3′, BoHV-1 R (reverse primer) 5′-GTA GTC GAG CAG ACC CGT GTC-3′ and a BoHV-1 probe (FAM-MGB) 5′-AGG ACC GCG AGT TCT TGC CGC-3′. The b*ACTB* TaqMan^®^ Custom Gene Expression assay (Applied Biosystems) comprised b*ACTB* F (forward primer) 5′-CCC TGG AGA AGA GCT ACG AG-3′, bACTB R (reverse primer) 5′-CAG GAA GGA AGG CTG GAA GA-3′ and a bACTB probe (FAM-MGB) 5′-CGG TTC CGC TGC CCT GAG GC-3′. For each qPCR reaction, 12.5 µL of 2X RT-PCR buffer (Thermo Fisher Scientific, Carlsbad, CA, USA), 2.5 µL of the TaqMan Custom Gene Expression Assay (Applied Biosystems, Waltham, MA, USA), 1 µL of 25X RT-PCR Enzyme Mix (Thermo Fisher Scientific, Carlsbad, CA, USA) and 1 µL of the sample were combined for a total reaction volume of 25 µL per well of a 96-well qPCR plate. qPCR reactions were run on a 96-well plate on an ABI7500 FAST machine (Applied Biosystems, Waltham, MA, USA) using the following PCR cycling conditions: 50 °C (2 min), 95 °C (10 min), and then 40 cycles of 95 °C (15 s) and 60 °C (1 min). Cq values were converted to relative quantities using 2^−(*UL27* Cq-b*ACTB* Cq)^. Cq values were converted to the number of BoHV-1 genome copies per µL and the number of *Bos taurus* genome copies per µL using two standard curves (qPCR Cq value vs. DNA concentration) generated from a dilution series of extracted *Bos taurus* genomic DNA and a PCR2.1 TOPO plasmid (Thermo Fisher Scientific, Carlsbad, CA, USA) clone of the BoHV-1 *UL27* gene that we prepared (according to the manufacturer’s instructions) and quantified using both spectrophotometry (on a Nanodrop 1000) and DNA specific dye fluorescence (on a Qubit fluorometer). The mean of the Nanodrop DNA concentration value and Qubit DNA concentration value was used for the generation of the standard curves.

### 2.7. Generation of MinION Nanopore Libraries for Multiplex Rapid Sequencing

MinION nanopore sequencing libraries were generated from nucleic acid extracted from bFLCs and nasal swab-2 eluates with the Rapid PCR Barcoding Kit (SQK-RPB004) [27] (Oxford Nanopore Technologies, Oxford, UK). For each library, 6 µL of nucleic acid extraction plus 2 µL of fragmentation mix (FRM) were added to a thin-walled 0.5 mL PCR tube (Eppendorf, Hamburg, Germany). The tubes were incubated in a Master Cycler Gradient PCR machine (Eppendorf, Hamburg, Germany) at 30 °C for 1 min and then 80 °C for 1 min, after which they were immediately placed on ice. This resulted in tagmentation of the DNA in the nucleic acid extraction with sequencing adapters. 

For PCR amplification of the tagmented DNA, 8 µL of the tagmentation reaction, 16 µL of nuclease-free water, 1 µL of rapid barcode primer (RLB), and either 25 µL of NEB LongAmp *Taq* 2× Master Mix (New England BioLabs Inc., Ipswich, MA, USA) or 25 µL of NEB Next Ultra II Q5 Master Mix (New England BioLabs Inc.) were added to a 0.5 mL thin-walled PCR tube (Eppendorf, Hamburg, Germany). The tube contents were mixed by gently flicking the tube and then centrifuged for 10 s in a minifuge. The tubes were then placed in a Master Cycler Gradient PCR machine (Eppendorf AG, Hamburg, Germany). Cycle conditions for LongAmp *Taq* were: 95 °C (3 min); then 10, 20 or 30 cycles of 95 °C (15 s), 56 °C (15 s) and 65 °C (6 min); and then one cycle of 65 °C (6 min) followed by a hold step at 4 °C. Cycle conditions for NEB Next Ultra™ II Q5 were 98 °C (3 min); then 10, 20 or 30 cycles of 98 °C (10 s), 65 °C (30 s) and 72 °C (either 40 s, 120 s, 180 s or 300 s); and then one cycle of 72 °C (2 min) followed by a hold step at 4 °C.

Following PCR, the 12 PCR libraries were combined in a single 1.5 mL Eppendorf DNA Lo-Bind (Eppendorf AG, Hamburg, Germany) tube and a total volume of 360 µL of AMPureXP beads (i.e., 30 µL of beads for each barcoded library) (Beckman Coulter Inc., Brea, CA, USA) was added to the 12 pooled PCR reactions and mixed gently via pipetting. The library pool and beads were incubated in a rotator mixer for 5 min at room temperature. The tube was then removed from the rotator mixer, centrifuged for approximately 20 s in a minifuge, placed on a magnet for 5 min and the supernatant was discarded. Being careful not to dislodge the beads, 1 mL of 70% ethanol was added to the beads, then immediately removed. This ethanol wash step was then repeated once. The tube was then centrifuged for 30 s using a minifuge and placed on the magnet until the beads had bound to the side of the tube, leaving the ethanol at the bottom. Residual ethanol was removed from the bottom of the tube via pipetting without touching the beads. The tube was then left open for 60 s to allow the pellet to air dry. The tube was then removed from the magnet and the pellet was resuspended in 25 µL of a solution of 10 mM Tris-HCl (pH 8.0) and 50 mM NaCl. The solution was mixed by pipetting until the pellet was completely resuspended and the suspension was then incubated for 2 min at room temperature to elute the purified–pooled–barcoded libraries. The tube was placed on the magnet for 5 min and the eluate containing the purified–pooled–barcoded libraries was collected and transferred to a new 1.5 mL DNA LoBind tube (Eppendorf AG, Hamburg, Germany). A volume of 1 µL of the purified–pooled–barcoded libraries was removed to measure its DNA concentration on a Qubit fluorometer using the Qubit 1× dsDNA HS Assay Kit (Thermo Fisher Scientific, Carlsbad, CA, USA), following the manufacturer’s instructions. The volume of the library corresponding to 60 ng dsDNA was calculated and removed to a new 1.5 mL DNA LoBind tube. This volume was then adjusted to 10 µL with 10 mM Tris-HCl (pH 8.0) and 50 mM NaCl. One microliter of rapid adapter (RAP) was then added to the 10 µL of the pooled barcoded library, mixed gently by flicking the tube, centrifuged for approximately 10 s in a minifuge to collect the liquid at the bottom of the tube and incubated at room temperature for 5 min. Subsequently, the sample and flowcell were prepared for priming and loading following the manufacturer’s instructions. The library was loaded into the spot-on port on a spot-on flowcell (FLO-MIN106D R9) (Oxford Nanopore Technologies, Oxford, UK) and sequenced for 48 h on a MinION Mk1b sequencer (Oxford Nanopore Technologies, Oxford, UK) attached to a MinIT compute module (Oxford Nanopore Technologies, Oxford, UK) using MinIT software version 19.05.02 (with Guppy 3.0.3) and the rapid basecalling option. Each flowcell was tested with the ONT configuration program in MinKnow software immediately prior to each run to ensure that the number of active pores was >1000. The output selected for each run was fast basecalling, FASTQ files only.

### 2.8. Epi2ME Analysis

FASTQ files were downloaded from the MinIT compute module to a 1 TB external hard drive on a laptop by connecting to the MinIT WiFi according to the manufacturer’s instructions. The FASTQ files were then uploaded to ‘Epi2ME Fastq WIMP Workflow’ or ‘Epi2ME Fastq Custom Alignment Workflow’ using the Epi2ME desktop agent installed on a laptop (Dell Latitude Precision 5520), using the default Epi2ME WIMP settings (i.e., no minimum or maximum read length and minimum quality score of 7). Default settings for the Epi2ME Custom Reference Alignment Workflow were also used. Genbank accession number AJ004801.1 was used as the BoHV-1 reference genome in all alignments.

### 2.9. Sequencing PCR-Free Libraries with the Field Sequencing Kit

Non-barcoded PCR-free libraries were generated using the Field Sequencing Library Preparation Kit (LRK001) (Oxford Nanopore Technologies, UK) according to the manufacturer’s instructions. The libraries were sequenced on a FLO-MIN106D R9.4.1 flowcell (one library per flowcell) on a MinION Mk1b sequencer, attached to a MinIT compute module, for 24 h (using the rapid base calling in ONT Minknow software that was installed on the MinIT) and the resulting FASTQ files were uploaded to the Epi2ME Fastq WIMP Workflow for taxonomic assignment.

### 2.10. MinION Sequencing of Nasal Swabs Calves Challenged with BoHV-1

Nasal swabs were collected daily from the six BoHV-1 challenged calves and six control (PBS challenged) calves from day −1 to day 6 relative to the challenge. Each swab was diluted in 3 mL of PBS, and a 1 mL aliquot of this was used for depleted nucleic acid extraction for sequencing and library preparation. Nucleic acid extraction and library preparation were performed in batches comprising the 8 nasal swabs that were collected from each animal (one swab per day) plus a clean swab as a negative control. To test the technical replication, in each batch, two libraries with different barcodes were prepared from the same swab for 3 of the 8 nucleic acid swab extracts. Each batch of 12 libraries was run on a separate flowcell. The libraries were sequenced on a FLO-MIN106D R9.4.1 flowcell on a MinION Mk1b sequencer attached to a MinIT compute module for 24 h (using rapid base calling in ONT Minknow software that was installed on the MinIT) and the resulting FASTQ files were uploaded to the Epi2ME Fastq WIMP Workflow for taxonomic assignment. 

### 2.11. Genome Assemblies

Adaptor sequences were removed from sequence reads using Porechop 0.2.4. Reads were aligned to the bovine genome using Minimap2 [28] (version 2.17-r974) to identify host-derived sequences. These were subsequently removed using SAMtools [29] (v 1.10). All reads less than 100 bp in length were removed using bbmap [30] (38.22) and the remaining reads were assembled using Flye [31] (2.8). Assembled contigs were further polished by aligning the original reads using Medaka (1.0.3) [32].

## 3. Results

### 3.1. Experimental Challenge of Calves with BoHV-1 and PBS

From the day prior to the challenge to the sixth day post-infection, daily rectal temperatures and clinical signs were recorded for each calf. The rectal temperatures increased from day 2 post-challenge in BoHV-1-challenged calves, whereas rectal temperatures did not increase in the PBS-challenged control calves (Figure 1). BoHV-1 qPCR analysis of the nasal swab-1 eluates (collected daily from every animal on all 8 days of the trial) showed that all swabs taken one day after the challenge with BoHV-1 were qPCR-positive for BoHV-1 with Cq values ranging from 22 to 35 (Appendix A). All nasal swab-1 eluates from swabs collected prior to the challenge or from animals that were challenged with PBS were qPCR-negative for BoHV-1 (Appendix A). 

### 3.2. qPCR Analysis of Bovine Nucleic Acid Depletion in BoHV-1-Infected bFLC Cultures

Initially, due to the limited availability of nasal swabs from calves infected with BoHV-1, the assessment of the non-viral nucleic acid depletion step was conducted using aliquots (of the same batch) of the in vitro BoHV-1-infected bFLC culture. These aliquots were subjected to five different combinations (A, B, C, D and E) of bead beating and nuclease treatment. Nucleic acids were then extracted and qPCRs of the *bACTB* gene and *UL27* gene were used to measure the relative quantities of bovine and BoHV-1 DNA respectively (Table 1). 

The most effective method identified for the depletion of non-viral nucleic acid was an initial bead-beating treatment, followed by a single treatment with RNaseA and then two 30 min incubations with Turbo DNase (treatment E). However, whilst treatment E (bead beating combined with single RNaseA and double DNase treatment) depletion resulted in the greatest reduction in bovine DNA, it also resulted in a 10-fold loss of BoHV-1 DNA. Bead beating was likely to be the cause of this loss, as the other treatment, which included bead beating (treatment B), also showed a large reduction (16-fold) in BoHV-1 DNA. These large losses could have been due to the exposure of non-capsid viral nucleic acid from infected bovine cells (i.e., virocells) to nucleases following disruption of the cells via bead beating. Bead beating may fragment some of the exposed viral DNA to sizes that are too small to be recovered by the QIAamp UltraSens Virus Kit or amplified by the *UL27* primers. In order to achieve high sensitivity of BoHV-1 detection on the MinION, it was necessary to increase the ratio of viral to non-viral nucleic acid, even if that meant losing BoHV-1 DNA.

### 3.3. Effect of Non-Viral Nucleic Acid Depletion on MinION Nanopore Sequencing

As expected, following Epi2ME Fastq WIMP analysis, the percentage of reads that were assigned to viruses was increased in the depleted cell culture libraries (mean = 96.94%, s.d. ± 0.12, n = 3) compared with undepleted cell culture libraries (mean = 45.61%, s.d. ± 0.82, n = 3) (Table 2). The percentage of classified reads that were assigned to BoHV-1 was also dramatically increased in the depleted nasal swab library (12.03%) compared with the non-depleted (0.32%) nasal swab library (Table 2). However, the undepleted cell culture libraries had higher viral read counts (mean = 22,626, s.d. ± 8888, n = 3) than the depleted cell culture libraries (mean = 4164, s.d. ± 1108, n = 3). Whereas the undepleted library prepared from the swab had lower viral read counts (657) than the depleted library made from the same swab (1903). 

We also used the Epi2ME Fastq Custom Reference Alignment Workflow to align the rapid basecalled FASTQ file from the depleted and undepleted swab libraries to the BoHV-1 reference genome. For the undepleted swab, the alignment showed the following: average alignment length = 4013 bases, number of aligned bases = 1,657,348, number of aligned reads = 414, total bases sequenced = 1800 Mb and percentage of total bases sequenced that aligned with the BoHV-1 genome = 0.092%. The depleted library alignment showed the following: average alignment length = 1443 bases, number of aligned bases = 3,003,635 and number of aligned reads = 2082. Therefore, there was a 1.8-fold increase in the number of bases and a 38-fold increase in the percentage of bases that aligned with the BoHV-1 genome in the depleted library compared with the undepleted library. Even coverage of the entire genome was observed in both depleted and undepleted libraries with fewer but longer reads in the undepleted swab library (Appendix A).

### 3.4. Optimisation of Low-Bias PCR Amplification of Tagmented Libraries

Due to the high cost of FLO-MIN106D R9.4.1 flowcells, we aimed to develop a protocol whereby an optimal number of samples could be processed on a single FLO-MIN106D R9.4.1 flowcell. In order to achieve this, the rapid PCR barcoding kit provided the simplest, most rapid option for library preparation of the depleted libraries. This kit involves the random insertion of partial adapters into ds DNA via tagmentation with a Tn5 transposase complex, followed by PCR amplification with barcoded primers that target the inserted adaptors. During pilot experiments, we observed that tagmented libraries generated with the Rapid PCR Barcoding Kit (SQK-RPB004, Oxford Nanopore Technologies, Oxford, UK) from nucleic acid extraction from high titre BoHV-1-infected lung cell cultures showed unexpectedly low read counts for BoHV-1 after 30 cycles of PCR using the NEB LongAmp *Taq* 2× Master Mix that was recommended by ONT. These samples showed high read counts for BoHV-1 in PCR-free libraries prepared with the PCR-free Field Sequencing Kit; therefore, we suspected that the low BoHV-1 read counts with the Rapid PCR Barcoding Kit were due to PCR bias, as the BoHV-1 genome has a very high GC content (72%). This prompted us to compare NEB LongAmp *Taq* 2× Master Mix with NEB Next Ultra II Q5 Master Mix to assess their PCR bias during library preparation and the consequent differences in BoHV-1 read counts. NEB LongAmp *Taq* 2× Master Mix is recommended by ONT for use with their Rapid PCR Barcoding Library Preparation Kit as it can amplify long templates and generate long reads. However, it is not optimised for amplifying templates with extreme GC content, such as the BoHV-1 genome. NEB Next Ultra II Q5 polymerase is optimised for amplification of templates with a wide range of GC content but not for long templates. Many viral genomes have unusually high or low GC content; therefore, the GC bias of polymerases could lead to certain viruses being underrepresented or missed altogether following metagenomic library amplification bias [33]. Libraries were generated with either NEB LongAmp *Taq* or NEB Next Ultra II Q5 using either 10, 20 or 30 PCR cycles, after which they were sequenced on the MinION. Libraries generated with NEB Next Ultra II Q5 polymerase showed a consistently high percentage (≥96% of Epi2ME Fastq WIMP-classified reads) of BoHV-1 sequence reads for 10, 20 and 30 PCR amplification cycles, whereas the libraries generated with NEB LongAmp *Taq* showed a dramatic reduction in the percentage of BoHV-1 sequence reads with increasing PCR cycle number and a dramatic increase in the percentage of bacterial sequence reads that were detected after 20 and 30 PCR cycles (Figure 2). This was most likely due to the positive amplification bias of the NEB LongAmp *Taq* towards the lower GC content of the bacterial genomes. Alignment of the sequences from these libraries to the complete BoHV-1 genome using the ‘Custom Fastq Alignment Workflow’ in Epi2ME was also conducted. This showed a huge increase in the number of bases and number of reads aligning to the BoHV-1 genome with increasing PCR cycle number with NEB Next Ultra II Q5 compared with NEB LongAmp *Taq* (Appendix A, Figure 2). The percentage identities and accuracies produced by the Epi2ME Fastq Custom Alignment Workflow were also higher with NEB Next Ultra II Q5 than with LongAmp *Taq* Appendix A. However, Q5 yielded shorter read alignments (average alignment length = 280.7 bases, s.d. ± 64.5) to BoHV-1 than NEB LongAmp *Taq* (average alignment length = 2304.5 bases, s.d. ± 2384.6) in these libraries where an extension time of 5 min was used for the Q5 PCR (Appendix A). It was later observed that a 40 s annealing time for Q5 led to longer alignments (>1 kb) (see below).

In terms of sequence read counts, the Q5 polymerase dramatically increased the numbers of BoHV-1 reads after 20 PCR cycles (109-fold increase) and 30 cycles (169-fold increase) (Table 3). After 30 PCR cycles with Q5, tens of thousands of BoHV-1 reads were identified using Epi2ME WIMP. With LongAmp *Taq*, there was a 0.3-fold decrease in the BoHV-1 read count between 10 and 20 cycles and only a 3.7-fold increase between 10 and 30 cycles with just a few hundred BoHV-1 reads after 30 cycles (Table 3). The LongAmp *Taq* showed a much higher amplification of the bacterial sequence than Q5. The amplification of eukaryotic nucleotide sequence was similar using both polymerases. 

We also tested whether it was possible to reduce the time of the PCR extension step for each PCR cycle with the Q5 polymerase. The 5-min extension step recommended by ONT for LongAmp *Taq* resulted in a PCR amplification step that took 4 h and 10 min. The extension times tested for Q5 were 5 min (as recommended by ONT), 3 min, 2 min and 40 s. Unexpectedly, the longest reads were obtained with 40 s of extension, with an average read length of 1400 bp. The reduction in the PCR extension step to 40 s led to a reduction in the overall 30-cycle PCR amplification to 80 min. This meant the entire protocol from swab to successful untargeted detection of BoHV-1 could be completed in 7 h in some of the nasal swabs where animals were shedding high amounts of virus (Figure 3).

### 3.5. MinION Sequencing of Nasal Swabs from Calves Challenged with BoHV-1 or PBS

BoHV-1 reads were identified using the Epi2ME Fastq WIMP Workflow in all the sequence libraries that were generated from nasal swabs taken from the BoHV-1 challenge group from day 1 to day 6 post-infection. BoHV-1 was not detected in any of the nasal swabs from calves challenged with PBS (Table 4). BoHV-1 was not detected on day −1 and 0 in four of the six calves challenged with BoHV-1 (Table 4). One or two reads were identified as BoHV-1 for day −1 for one calf from the BoHV-1 challenge group, and for day 0 for two calves from the BoHV-1 challenge group (Table 4). One or two reads were also identified as BoHV-1 in the negative extraction control that was included in the batch of extractions from those same two BoHV-1-challenged animals (BoHV1_1 and BoHV1_2) (Table 4). The read counts and percentages of reads (percentage classified and percentage classified plus unclassified) for non-BoHV-1 viruses, eukaryotes, bacteria and archaea are shown in Appendix A.

### 3.6. BoHV-1 Sequence Yield Barcode Variation

In terms of the amount of the BoHV-1 sequence generated from swabs from infected animals, as much as 223.6 Mb of BoHV-1 sequence was produced from a single swab on one of 12 barcodes on the calf BoHV1_1 flowcell, but the BoHV-1 sequence yield and coverage of the BoHV-1 genome varied considerably between swabs and even between barcodes where two different barcodes were used for duplicate libraries generated from the same swab nucleic acid extraction (Table 5). There was also wide variation between the BoHV-1 read numbers between samples with different barcodes (Table 5). The alignments showed that there was preferential amplification of two parts of the BoHV-1 genome at nucleotide positions 20–25 kb and 118–122 kb (Figure 4); therefore, further work is required to achieve more uniform sequence coverage of the BoHV-1 genome. 

There was greater consistency between the duplicate libraries made from the same nucleic acid extraction from swabs taken on day 1 (barcode 4 and 5) and day 2 (barcodes 6 and 7) of the viral challenge when the number of bases that aligned with the BoHV-1 genome were calculated as a percentage of all bases sequenced (Table 5, Appendix A). 

### 3.7. Relationship between Sequencing and qPCR

As qPCR analysis of undepleted swabs is currently the standard method for the analysis of viruses in BRD cases, we assessed whether there was any relationship between the qPCR analysis of BoHV-1 and (i) the BoHV-1 read counts from Epi2ME Fastq WIMP analysis and (ii) the percentage of reads that aligned with the BoHV-1 reference genome following Epi2ME Fastq Custom Alignment Workflow analysis (Appendix A). For this, we conducted qPCR for the absolute and relative quantification of BoHV-1 using primers that targeted the BoHV-1 *UL27* gene and the *Bos taurus* b*ACTB* gene on nucleic acid extracted from the same nasal swab-2 eluates from calf BoHV1_1 nasal swabs that were used for the sequence analysis. XY scatterplot analysis (conducted in Microsoft Excel) showed that there was only a weak positive linear relationship between the BoHV-1 read counts and the relative and the absolute copy number qPCR of BoHV-1 in the depleted (absolute copy number qPCR R^2^ = 0.56, relative qPCR R^2^ = 0.57) and undepleted (absolute qPCR R^2^ = 0.53, relative qPCR R^2^ = 0.57) swab eluates. There was a strong positive linear relationship between the percentage of reads that aligned with the BoHV-1 reference genome and relative qPCR quantities of BoHV-1 in the depleted (R^2^ = 0.97) and undepleted (R^2^ = 0.97) swab eluates. The relationship between the absolute qPCR BoHV-1 copy number and the percentage of bases that aligned to the BoHV-1 reference genome was not as strong (depleted R^2^ = 0.92, undepleted R^2^ = 0.83).

### 3.8. BoHV-1 Sequence Assembly Direct from Swabs

To determine whether the sequences produced from swabs from a single animal run on a single FLO-MIN106D R9.4.1 flowcell could be used to assemble a viral genome with rapid basecalled FASTQ files, we performed untargeted sequence assembly (excluding bovine sequence) from all the rapid basecalled FASTQ files generated from the barcoded libraries from the calf BoHV1_1 flowcell that are shown in Table 5. Five major contigs were generated (NCBI GenBank accession numbers: BankIt2554305 contig_1_segment0 OM860299, BankIt2554305 contig_2_segment0 OM860300, BankIt2554305 contig_3_segment0 OM860301, BankIt2554305 contig_4_segment0 OM860302 and BankIt2554305 contig_5_segment0 OM860303), which were all identified as BoHV-1. These contigs covered approximately 60% of the BoHV-1 genome. The length, percent identity (%ID) and percent query cover (% cover) relative to the top hit following an nr/nt BLAST search were: contig 1 = 23,959 nt (99.79% ID, 100% cover), contig 2 = 16,595 nt (99.62%, 93% cover), contig 3 = 9547 nt (99.75% ID, 87% cover), contig 4 = 16,237 nt (99.82 %ID, 100% cover) and contig 5 = 13,703 nt (99.79%, 100% cover). The top 20 BLAST nr/nt hits were BoHV-1 complete genomes for all five contigs.

### 3.9. Detection of Viruses Other Than BoHV-1 in Nasal Swabs from BoHV-1 Calf Challenge Model

For each swab, many single reads were assigned to viruses/phages other than BoHV-1, including bacteriophages and eukaryotic viruses (Appendix A). Where one or two reads were assigned to each of these eukaryotic viral taxa, they were possibly a result of the incorrect assignment of bovine, fungal or yeast genomes, which were present in large amounts in these nasal swabs, or inaccurate submissions to the RefSeq database employed by WIMP. Incorrect assignments in WIMP due to inaccurate submissions to RefSeq were reported previously [34]. In the experimentally challenged calf BoHV1_1, the average assignment scores after Q7 filtering for BoHV-1 were 7145 (n = 77,958, s.d. 6282.4), whereas these scores were only 3152 for BoHV-5 (n = 1163, s.d. 2815.5) and 1494 for BuHV-1 (n = 153, s.d. 2234.6). To assess whether the incorrect assignments were due to read quality scores, we compared the Epi2ME analyses with the read quality scores >Q7 and >Q10 for animal BoHV1_1 (Appendix A). At >Q10, there were far fewer viral taxonomic assignments overall, the percentage of BoHV-1-assigned reads increased from 95.77% to 98.36%, and the percentage of non-viral taxa was reduced from 4.23% at Q7 to 1.64% at Q10. A 2.6-fold reduction in the percentage of incorrect assignments was found at Q10 vs. Q7. The day 0 swab sample, where BoHV-1 reads were not expected, showed a single BoHV-1-assigned read at Q7 and no BoHV-1-assigned reads at Q10. However, a 5.9-fold reduction in BoHV-1 detection sensitivity was found when Q10 quality score filtering was applied to Epi2ME Fastq WIMP Workflow pathogen identification software (92,149 and 15,631 reads were assigned to BoHV-1 with Q7 and Q10 filtering, respectively). 

There were many bacteriophage taxonomic assignments made following the Epi2ME Fastq WIMP analysis. More than 100 reads were assigned to Proteus phage VB_PmiS-Isfahan, Acinetobacter phage YMC13/03/R2096, bubaline alphaherpesvirus-1 (BuHV-1) and bovine alphaherpes virus-5 (BoHV-5) in some swabs. Reads that were assigned to the alphaherpes virus taxa other than BoHV-1 (e.g., BuHV-1 and BoHV-5) only occurred in animals that were challenged with BoHV-1, indicating that they were incorrectly assigned BoHV-1 sequences. The ratio of the non-BoHV-1 herpes virus assignments relative to BoHV-1 assignments also decreased when >Q10 filtering was used for the Epi2ME analysis (Appendix A).

Following submission to the NCBI SRA, an alternative taxonomic analysis of these FASTQ sequences was conducted by NCBI using STAT [35]. These NCBI STAT analyses showed far fewer non-BoHV-1 viral sequences than the Epi2ME Fastq WIMP analyses. For example, the STAT analysis conducted by NCBI on the calf BoHV1_1, day 2, barcode 7 FASTQ files did not detect any non-alpha herpes viruses. This indicated that with the parameters we selected, there was a higher level of assignment of these FASTQ files to incorrect taxa with Epi2ME Fastq WIMP than with STAT. NCBI STAT taxonomic analyses of all nanopore FASTQ files from this work are also available to view in the NCBI SRA run selector using project number PRJNA783212.

### 3.10. Incorrect Assignment of Bos taurus Sequence to Viral and Bacterial Taxa by Epi2ME Fastq WIMP

The detection of Proteus phage VB_PmiS-Isfahan in bovine nasal swabs by Epi2ME Fastq WIMP was unexpected and was not observed in any of the nasal swab samples following the STAT analysis (see STAT-assigned taxa in the NCBI SRA run selector using project number PRJNA783212, Appendix A). Five reads (all longer than 2 kb) that had been assigned by Epi2ME to the taxon ‘Proteus phage VB_PmiS-Isfahan’ were randomly selected and subjected to NCBI BLAST nr/nt analysis (Appendix A). The top 100 hits for all five of these reads were *Bos taurus*, indicating that bovine sequence from the nasal swabs had been incorrectly assigned to viral taxa by Epi2ME Fastq WIMP. This indicated that the Proteus phage VB_PmiS-Isfahan taxon in the Epi2ME WIMP RefSeq database, which is a single complete genome assembly (RefSeq accession NC_041925.1), is contaminated with a bovine sequence. This was reported to NCBI and they subsequently removed this contaminated assembly from their database.

The analysis of sequences by Epi2ME Fastq WIMP Workflow pathogen identification software using the Epi2ME desktop agent after 2021 unexpectedly showed high levels of *Clostridium botulinum*. In 2020, *Clostridium botulinum* was not detected with this software in the swabs from these experimentally challenged calves. In March 2021, we reanalysed the FASTQ files from animal BoHV1_1 that had been previously analysed in February 2020. *Clostridium botulinum* was not detected when Epi2ME Fastq WIMP analysis of these FASTQ files was conducted in February 2020. However, *Clostridium botulinum* was detected at high levels when Epi2ME Fastq WIMP analysis of these same FASTQ files was conducted in March 2021 (Appendix A). The Epi2ME Fastq WIMP taxonomy reports showed that many sequences were being assigned to RefSeq sequence accession NZ_CP027778.1 (*Clostridium botulinum* strain Mfbjulcb6 chromosome, complete genome 2018). STAT analysis of nanopore FASTQ files was performed by NCBI from the nasal swabs and did not show that *Clostridium botulinum* was present in any of the swabs from the BoHV-1 challenge. NCBI investigated the sequence and determined that this accession was contaminated with five bovine segments, as well as yeast and plant contamination. However, this contaminated assembly remained in the list of 56,044 sequence accessions from RefSeq that was employed by Epi2ME Fastq WIMP. Consequently, Epi2ME Fastq WIMP currently incorrectly assigns high numbers of bovine sequence reads to this *Clostridium botulinum* accession.

## 4. Discussion

The current work demonstrated that correct same-day detection of a known virus in nasal swabs of experimentally challenged calves was achievable using the ONT MinION and its associated Epi2ME cloud-based software. This involved subjecting the nasal swab eluate to mechanical disruption by bead beating, nuclease depletion of non-viral capsid nucleic acid, simple tagmentation-based library preparation with PCR barcoding (to allow multiplexing of libraries, generated from samples and controls, on a single flowcell), rapid base calling of MinION Nanopore sequenceon a MinIT compute module, and rapid cloud-based Epi2ME Fastq WIMP Workflow pathogen identification software.

Due to the fact it was developed prior to the other two MinION flowcells, most MinION sequencing protocols, including rapid tagmentation library preps, were developed using FLO-MIN106D (R9.4.1), which typically has 1200–1600 active R9.4.1 pores. FLO-FLG001 (R9.4.1) is one-tenth of the cost but has only 80–160 active pores. FLO-MIN112 (R10.4) has higher consensus sequencing accuracy but is currently only optimised for libraries prepared using relatively slow ligation methods [36]. With the FLO-MIN106D (R9.4.1), using the rapid basecalling default option on ONT MinKnow sequencing software, a file containing 4000 basecalled FASTQ reads is generated approximately every two minutes in the early stages of the sequence run. The rate of sequencing declines as the sequence run progresses. 

As soon as these FASTQ files are generated, they can be uploaded to a cloud-based software platform called Epi2ME, which contains several intuitive point-and-click sequence analysis applications called ‘workflows’. In the present work, the viral, bacterial, fungal and yeast sequences were identified from nanopore FASTQ files using the ‘Epi2ME Fastq WIMP Workflow’, which employed the ‘Centrifuge’ algorithm [37]. Centrifuge uses an indexing scheme based on the Burrows–Wheeler transform (BWT) and Ferragina–Manzini (FM) index, which were optimised specifically for metagenomic classification. Centrifuge has space-optimised indexing schemes, requires a relatively small index and classifies sequences at a very high speed; therefore, it can process the millions of sequence reads from a typical high-throughput DNA sequencing run within a few minutes on a desktop computer or laptop [37]. As long as the proportion of pathogen nucleic acid in the samples relative to that of non-pathogen nucleic acid is sufficiently high, Epi2ME Fastq WIMP Workflow enables the identification of a pathogen within approximately 15 to 30 min of loading a sequencing library on a FLO-MIN106D (R9.4.1) flow cell.

The size of most viral genomes is several orders of magnitude lower than those of bacteria and eukaryotes. Consequently, in nasal swabs taken from cattle infected with a BRD-associated virus, the vast majority of the nucleic acid will be prokaryotic and eukaryotic, and just a fraction will be viral. Several methods are commonly employed to enrich the viral sequence relative to the non-viral sequence in a sample to decrease the amount of sequence depth required to obtain a viral genome sequence from complex samples. The ViroCap targeted sequence capture panel was designed to enrich the nucleic acid from DNA and RNA viruses from 34 families that infect vertebrate hosts [38] but can detect many viruses that are not on the panel [39]. ViroCap was used to enrich the animal viruses from clinical samples for sequencing on the MinION but the optimal probe hybridistaion time varied for different viruses and a 20 h probe hybridisation time was adopted [40]. PCR amplification (using overlapping targeted primers spanning an entire viral genome (spiked primer approach) can be used to increase the amount of whole viral genome sequence from a sample if the genome sequence of the virus is known [41].

Enrichment of viral nucleic acid can also be achieved via the depletion of non-viral material from a sample. Eukaryotic and prokaryote cells can be separated from the much smaller viral capsids via ultracentrifugation [42]. However, some giant viruses, such as mimiviruses, which are associated with pneumonia in humans, are larger than some bacteria and thus pellet at lower centrifugation speeds than bacteria [43]. 

As intact viral capsids are nuclease resistant, RNaseA and DNase1 can be used to selectively digest non-viral capsid nucleic acids. DNase1 and RNaseA are applied following cell disruption so that the eukaryotic and prokaryotic nucleic acids are exposed to the nucleases [44]. However, in a cell infected with a virus (i.e., a virocell), much of the virus nucleic acid is not protected by a capsid and this unprotected viral sequence can also be lost if cell disruption and nuclease pre-nucleic acid extraction treatments are applied.

Following the depletion of non-viral nucleic acid, there is often insufficient total nucleic acid to generate enough of a sequencing library for NGS and TGS platforms; therefore, following double-stranded cDNA synthesis, whole-genome amplification (WGA) approaches are usually applied to amplify all of the remaining total nucleic acids in a depleted nucleic acid preparation. These approaches include Sequence-Independent, Single-Primer Amplification (SISPA) and Linker Amplified Shotgun Library (LASL) [32,33,42,45], which both employ PCR, and isothermal multiple displacement amplification (MDA) using podovirus φ29 polymerase [42]. Not surprisingly, each WGA method was shown to preferentially amplify different families of viruses and MDA is prone to the generation of a chimeric sequence [45]. 

The protocol we developed employed the LASL WGA approach. Compared with MDA and SISPA, LASL sequencing requires fewer reagents, thus lower cost, and fewer steps, thus less time from taking the sample to loading the flowcell. With the LASL procedure we developed, the addition of library adapters and WGA simply comprises a 5-min tagmentation of nucleic acid with a sequencing adapter, followed by a 80-min, 30-cycle PCR amplification with barcoded primers.

There is currently a paucity of literature that describes the use of the experimental challenge of cattle with a known BRD virus to assess these relatively new viral metagenomics approaches. Nanopore sequencing has been used and will likely be increasingly used due to its many advantages over other next-generation sequencing platforms to compare the nasal viromes of cattle with and without BRD to attempt to find or confirm the associations of viruses with BRD [19]. However, nanopore viral metagenomics should be assessed for several BRD-associated viruses using experimental challenges in cattle with known viruses to check for sensitivity and specificity issues caused by extractability from swabs, varying GC content and varying amounts of extractable viral nucleic acid in swabs from the same animal during infection. 

Another group reported the assessment of nanopore sequencing of nasal swabs and tracheal washes from animals that were identified as infected with a BRD-associated virus [46]. Rather than using experimentally challenged animals, they screened nasal swabs and tracheal washes from 116 animals using qPCR and MiSeq and found that 19 samples were naturally infected with the influenza D virus (IDV). They performed nuclease depletion (with DNase and RNase) of non-capsid-protected nuclease acid in the IDV-positive nasal swab and tracheal wash samples. However, they did not report cell disruption prior to the nuclease treatment. They used a random primer ‘FR20RV’ for WGS but did not give details of the polymerase or cycling conditions; therefore, it is not possible to know whether their protocol would have been able to sequence large viral genomes or genomes with extreme GC content directly from samples. Unlike BoHV-1, which has a relatively large genome (135.3 kb) with very high GC content (72%), IDV only has a small genome (12.3 kb) with an average GC content (41.5%). They generated libraries with the Ligation 1D Sequencing Kit SQK-LSK108, and thus, library generation would have been considerably slower with many more pipetting steps than our LASL method, although they did not report the time it took from sample to result. They also ran the libraries on a GridION, not a MinION. The GridION uses the same flowcells and similar software to the MinION but, unlike the MinION, it is not portable or low-cost. 

By looking at a single ‘known’ BRD virus in infected nasal swabs and cell cultures, we were able to reveal significant technical issues with the standard ONT protocols. One of the major problems we encountered was that LongAmp *Taq* failed to amplify the BoHV-1 genome efficiently and, instead, preferentially amplified the non-viral DNA. NEB LongAmp *Taq* is suggested by ONT for use in many of their protocols, including the whole-genome amplification of nanopore libraries. The very high GC content of the BoHV-1 genomewas most likely the cause of the failure of the PCR amplification of BoHV-1 DNA with NEB LongAmp *Taq*. We showed that NEB Next Ultra II Q5 polymerase gave far higher PCR amplification of BoHV-1 DNA than NEB LongAmp *Taq*. Whether NEB Next Ultra II Q5 polymerase gives representative amplification of all viruses in the cattle nasal virome (including dsDNA, ssRNA, dsRNA and ssDNA viruses with high, low or average GC content) will have to be carefully assessed in nasal swabs from experimental challenge models in cattle with a range of viruses. A reverse transcription step will also have to be optimised for RNA viruses, as different reverse transcriptases vary widely in their performance in achieving optimal sequence coverage of RNA viral genomes [47]. Nevertheless, we demonstrated that, with the introduction of double-stranded cDNA synthesis, a rapid tagmentation-based nanopore viral shotgun metagenomics approach could simultaneously and correctly detect RNA and DNA viruses in control mixtures of cultures of three BRD-associated viruses (BoHV-1, BPI3 and BRSV) [48,49]. We also recently showed that the current procedure can detect RNA and DNA viruses (e.g., bovine coronavirus, bovine rhinitis virus and ungulate tetraparvovirus) in nasal swabs from naturally infected animals [49].

Pooling and mixing swabs from different challenge models would also allow us to test the performance of the MinION sequencing and Epi2ME analysis with a ‘known’ mixture of viruses in infected nasal swabs. Most mock communities for viruses are generated from cell cultures. Cell cultures have far fewer non-viral nucleic acids than nasal swabs; therefore, they are not representative of nasal swabs and are consequently suboptimal for developing nasal swab sequencing protocols. In the current study, we observed a much lower percentage of viral sequence in both the depleted and undepleted libraries derived from nasal swabs than from cell cultures. A mock BRD RNA/DNA virus community from infected nasal swabs from experimentally challenged cattle models would be extremely useful for the assessment and optimisation of nanopore nasal virome sequencing protocols.

Spike-in controls would also allow for the determination of specificity. One or two reads were assigned to BoHV-1 in day −1 and day 0 samples and blank swab/PBS negative extraction controls in the first two of the six batches of swabs we processed from BoHV-1-challenged animals. This could have resulted from cross-contamination during DNA extraction or library preparation, and/or index hopping during sequencing. ONT sequencing was reported to generate 0.02–0.3% index hopping [50]. In the current work, stringent measures were adopted to avoid cross-contamination. To prevent sample aerosol escape during the high-speed bead beating and centrifugation, the lids of the screw-caps (with o-ring gaskets) pathogen lysis tubes were screwed tight and wrapped with Parafilm for bead beating and high-quality Eppendorf Safelock microfuge tubes were used for all centrifugation steps. Minimisation of manual handling steps of the samples could further reduce possible cross-contamination. Automated sample extraction and library preparation using devices such as the Voltrax [51] or PDQEX [52] would eliminate many possible cross-contamination steps, although this would require further optimisation. Index hopping could be reduced or eliminated via improved removal of unligated adapters and improved index sequences in the adapters that are supplied in the rapid PCR-barcoding kit.

It would be useful if the quantity of virus in the nasal swab could be estimated from the viral shotgun metagenomics sequence data. However, in the current work, the qPCR analysis showed that BoHV-1 was greatly reduced in depleted, compared with undepleted, nasal swab eluates. Adding an accurately quantified spike-in control of a cocktail of different viruses immediately prior to extraction could allow for an estimation of the loss of viral nucleic acids due to depletion, thus enabling an estimation of the quantity of the virus in the sample prior to depletion. The addition of a second nucleic acid spike-in control immediately prior to adapter ligation and WGA library generation would also be necessary to control for the effects of library preparation and would allow for the estimation of viral quantities in nasal swab samples [53]. Adding spike-ins introduces the risk of cross-contamination of the viral sequence in the sample being analysed with the spike-in sequence [54]. Therefore, again, this would have to be carefully tested to ensure that the spike-ins are sufficiently different from the virus sequences in the sample being analysed. As we were still assessing which virus sequences were common to BRD nasal swabs in Ireland, we decided not to use spike-ins in the current work.

Although spike-ins would be necessary to allow for an estimation of the absolute amounts of virus from sequencing, the relative abundance between samples can be estimated without spike-ins. The only quantitative output from Epi2Me Fastq WIMP analysis is read counts. We found that the Epi2Me Fastq WIMP BoHV-1 read counts were very different when we ran the same extracted nucleic acid sample with two different barcodes on the same flowcell. Therefore, read counts (assigned by Epi2ME Fastq WIMP) for a particular virus cannot be relied on for comparing the relative abundance of that virus between samples. However, there was reasonable consistency between the same nucleic acid extract run with different barcodes when the Epi2ME Fastq Custom Alignment Workflow was used to calculate the percentage of bases that aligned with the BoHV-1 reference genome. The percentage of bases that aligned with the BoHV-1 reference genome also showed a stronger relationship with the qPCR analysis of BoHV-1 in the nasal swab eluates. Therefore, for quantification purposes, the Epi2ME Fastq Custom Alignment Workflow is more useful than the Epi2ME Fastq WIMP workflow.

Incorrect assignments of bovine nanopore sequence reads to viral and bacterial taxa by Epi2ME Fastq WIMP is a serious issue. The assignment of bovine sequence to *Clostridium botulinum* taxa could lead to the application of inappropriate, unnecessary and costly treatment and prevention measures on a farm. The incorrect assignment of a bovine sequence to Proteus phage VB_PmiS-Isfahan is also highly misleading in a research environment and the required time and effort to investigate and discount this as being present in the cattle upper respiratory tract was substantial. These instances were both due to the low-quality assemblies being released in RefSeq. Epi2ME Fastq WIMP cannot currently be relied on in a real-world situation and requires either a database with better curation than NCBI RefSeq or for NCBI to scrutinise submissions more carefully before releasing them onto the RefSeq database. There are likely many more low-quality assemblies in the 56,044 RefSeq sequences used by Epi2ME Fastq WIMP, as we observed that many unexpected, i.e., non-herpes viruses, were detected by this software and not by STAT. However, we did not investigate all of these to determine whether they were incorrectly assigned bovine sequences.

The cultivation of viruses is slow, biased and challenging, and the vast majority of viruses remain uncultivated to date [55]. Therefore, another objective of using viral shotgun metagenomics on the BoHV-1 nasal swabs was to assess whether the relatively large viral genome of BoHV-1 could be assembled directly from nanopore sequence from swabs from an infected animal. In a previous report, IDV genomes were assembled directly from BRD nasal swab nanopore sequences [46]. The largest IDV contigs assembled for each sample from Nanopore data ranged from 626 bases to 2308 bases [46]. The length of our BoHV-1 contigs ranged from 9547 bases to 23,959 bases and we were able to assemble 60% of the BoHV-1 genome directly from nasal swab nanopore sequence taken from a single BRD case. Therefore, depending on the viral load, it should possible to obtain a sequence of a new/unknown virus directly from a nasal swab sample within 24–48 h. However, obtaining swabs with high enough levels of virus to allow a full genome assembly of a DNA virus with a genome >100 kb presents technical difficulties if swabs are taken from animals only after they show symptoms, such as high temperature. In four of the six BoHV-1 challenged calves, the highest number of BoHV-1 nanopore sequence reads were obtained in the swabs taken on days 1 and 2 post-BoHV-1 challenge. This was before the increases in rectal temperatures, which were observed on days 3, 4, 5 and 6 post-BoHV-1 challenge. In a BRD outbreak, it would be necessary to take swabs from a group of symptomatic and close-contact symptom-free animals over several days in order to acquire pre-symptomatic swabs with high enough viral loads to generate high-quality viral genome assemblies directly from nasal swab nanopore sequences.

For the viral genome assembly, we had to use third-party software on a local server, as Epi2ME so far only has an assembly workflow for SARS-CoV-2. While the Epi2ME software platform has huge potential for rapid user-friendly pathogen diagnostics, a major shortfall is the lack of an automated workflow for whole viral genome assembly from FAST5 or FASTQ files generated on the MinION from viral shotgun metagenomic sequencing. The addition of a rapid ‘Epi2ME Fastq viral shotgun metagenomics de novo genome assembly workflow’ to the Epi2ME suite of nanopore sequence analysis software would be extremely valuable for improving the diagnostics of BRD and ultimately the reduction of the incidence of this costly disease and its knock-on effects, particularly the spread of antibiotic resistance caused by the BRD-associated high-level use of broad-spectrum antibiotics.

## 5. Conclusions

Considerable optimisation and modification of the standard ONT library preparation protocols were necessary in order to achieve a protocol with sufficient sensitivity and speed to enable detection and generation of a genomic sequence of BoHV-1 virus in nasal swabs from BoHV-1-infected cattle within 24 h of swab collection using the ONT MinION device and Epi2ME cloud-based software. Of particular note, the NEB LongAmpTaq in the standard ONT rapid PCR barcoding library preparation protocol presented significant problems due to the preferential amplification of non-BoHV-1 DNA. We found this was resolved by changing to NEBNext Ultra II Q5 Master Mix. The accuracy and specificity of the WIMP workflow on the Epi2ME website require significant improvement before it can be used for clinical diagnosis. This is largely due to inaccurate whole-genome assemblies on the RefSeq database that Epi2ME WIMP utilises.

## Figures and Tables

**Figure 1 viruses-14-01859-f001:**
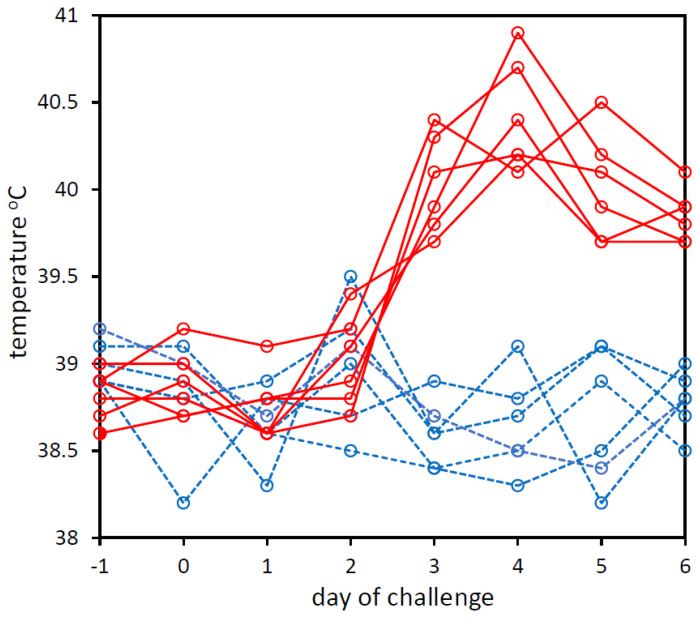
Rectal temperatures of calves that were experimentally challenged with PBS (dashed blue line) or BoHV-1 (solid red line). Temperatures on day −1 pre-challenge to day 6 post-challenge are shown. The graph was created in Microsoft Excel and Microsoft PowerPoint in Microsoft Office Professional Plus 2016.

**Figure 2 viruses-14-01859-f002:**
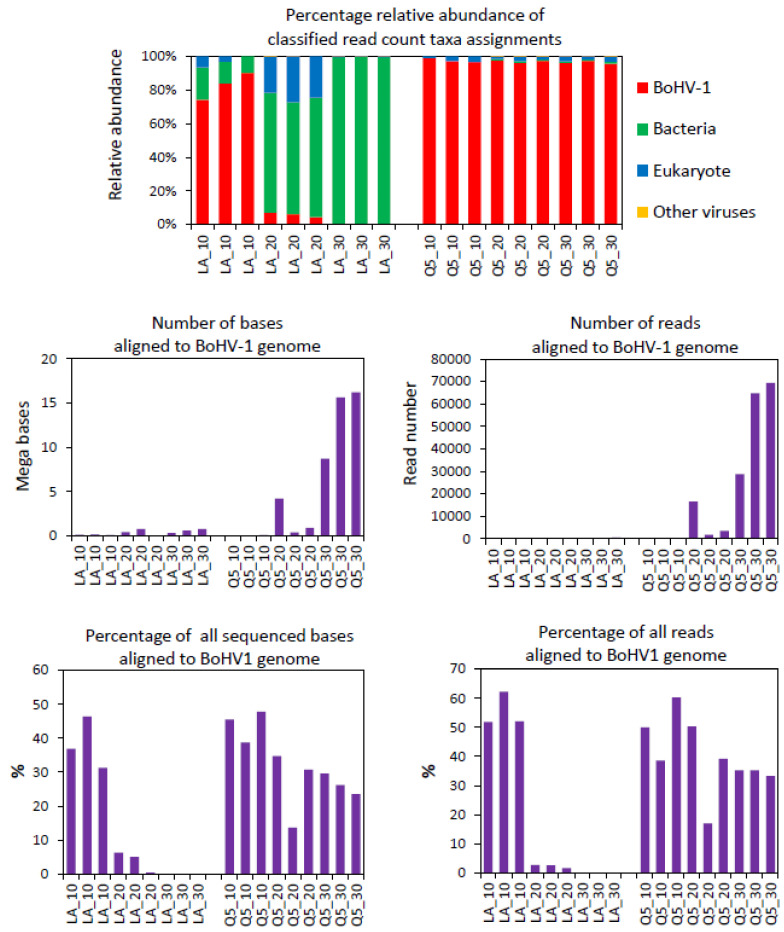
Comparison of BoHV-1 sequences generated from MinION sequencing libraries that were PCR amplified for 10, 20 or 30 cycles with either NEB Next Ultra II Q5 Master Mix (Q5) or NEB LongAmp *Taq* 2× Master Mix (LA) polymerases. Percentage read counts for taxonomic assignments were generated using Epi2ME Fastq WIMP Workflow analysis of FASTQ files. The number and percentage of aligned bases and reads were generated via alignment (using Epi2ME Fastq Custom Alignment) to the bovine herpesvirus type 1.1 complete reference genome (Genbank accession number AJ004801.1). Libraries were generated from the same nucleic extract (bead beating + nuclease) from BoHV-1 infected bFLCs using the ONT rapid PCR barcoding kit. Three libraries were generated for each of the 3 PCR cycle numbers. The graphs were created in Microsoft Excel and Microsoft PowerPoint in Microsoft Office Professional Plus 2016.

**Figure 3 viruses-14-01859-f003:**
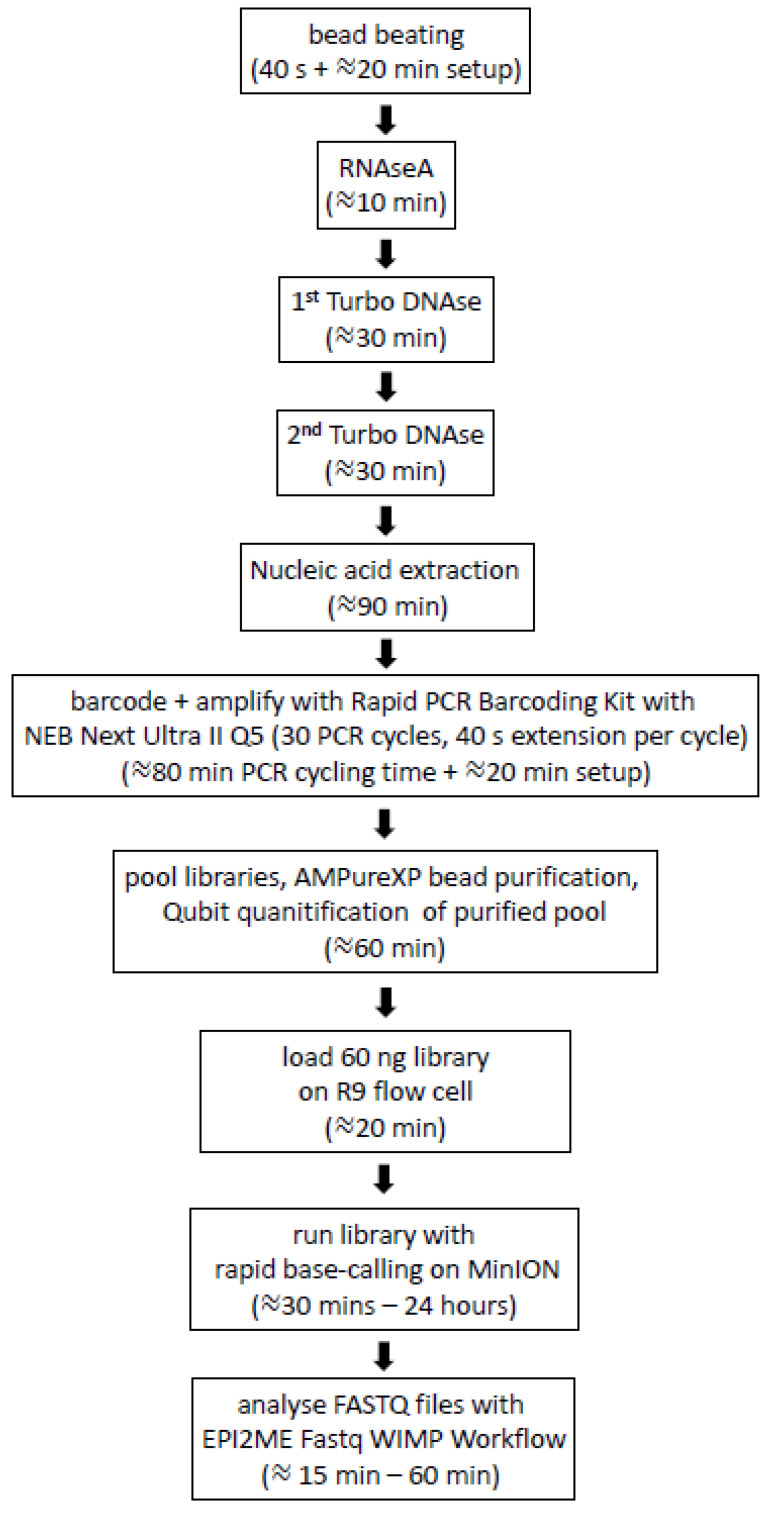
Outline of the protocol for the untargeted analysis of nasal DNA virome from nasal swabs of cattle using nanopore sequencing. Times indicated are from one person processing 12 samples in our laboratory (including a negative control). The image was created in Microsoft PowerPoint in Microsoft Office Professional Plus 2016.

**Figure 4 viruses-14-01859-f004:**
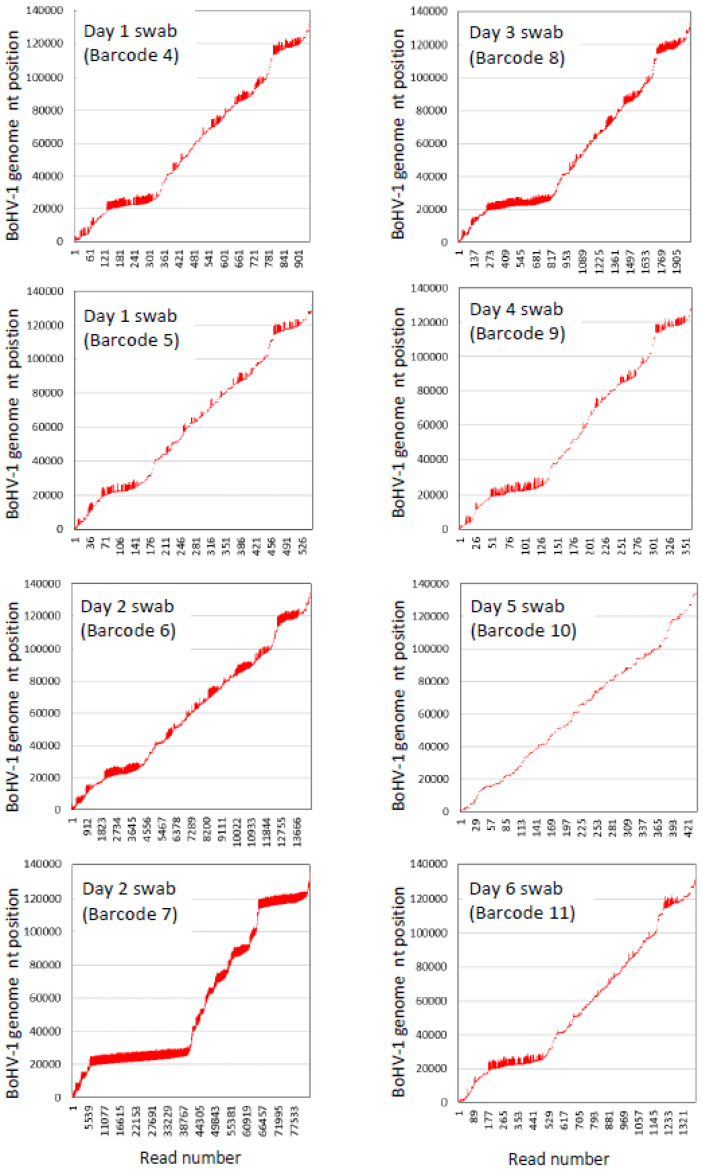
Epi2ME custom Fastq (Minimap 2) alignments of nanopore reads (represented by vertical red lines) from nasal swabs (one nucleic acid extraction per swab) from calf BoHV1_1 with the bovine herpesvirus type 1.1 complete genome sequence (GenBank accession number AJ004801.1).

**Table 1 viruses-14-01859-t001:** Effect of different non-viral nucleic acid depletion treatments on recovery of BoHV-1 and bovine genomic DNA from in vitro bFLC cultures.

Treatment	BoHV-1 Cq (*UL27*)	s.d.	Bovine Cq (*bACTB*)	s.d.	BoHV-1 (*UL27*) Relative Quantities	Bovine (b*ACTB*) Relative Quantities
A	20.76	0.09	32.00	0.21	1.00	1.00
B	24.79	0.40	35.72	0.67	−16.37	−13.20
C	20.60	0.54	33.80	1.75	1.11	−3.48
D	21.70	0.15	35.20	0.92	−1.92	−9.19
E	24.04	0.37	38.33	1.49	−9.77	−80.63

Note: mean qPCR quantitative cycle (Cq) values (n = 3) are shown for the three extraction treatment replicates, which were analysed using qPCR of the BoHV-1 *UL27* gene and the bovine bACTB gene. s.d.—standard deviation. Different treatments are indicated as (A) no depletion; (B) bead beating only; (C) 1× RNase and 1× DNase; (D) 1× RNase and 2× DNase; and (E) bead beating, 1× RNase and 2× DNase.

**Table 2 viruses-14-01859-t002:** Comparison of the read counts assigned to different taxa in undepleted and depleted libraries.

Taxonomic Assignment	Undepleted				Depleted			
	bFLC1	bFLC2	bFLC3	Swab	bFLC1	bFLC2	bFLC3	Swab
	Number of Reads Assigned to BoHV-1							
Eukaryota	>31,850	14,427	33,459	154,002	75	54	51	12,541
BoHV-1	24,295	11,854	27,180	501	4941	3339	3059	1781
Other viruses	1868	660	2022	156	492	338	325	122
Bacteria	528	252	563	2057	90	62	60	279
Archaea	31	11	31	126	0	0	0	13
Classified	58,759	27,289	63,480	157,738	5604	3797	3502	14,799
Unclassified	246,018	75,995	253,937	341,679	7555	5794	5280	229,354
	Percentage of classified reads assigned to BoHV-1							
Eukaryota	54.38	53.03	52.90	98.19	1.34	1.42	1.46	85.10
BoHV-1	41.35	43.44	42.82	0.32	88.17	87.94	87.35	12.03
Other viruses	3.18	2.42	3.19	0.10	8.78	8.90	9.28	0.82
Bacteria	0.90	0.93	0.89	1.31	1.61	1.63	1.72	1.89
Archaea	0.05	0.04	0.05	0.08	0.00	0.00	0.00	0.09
	Percentage of classified and unclassified reads assigned to BoHV-1							
Eukaryota	10.45	13.97	10.54	30.84	0.57	0.56	0.58	5.14
BoHV-1	7.97	11.48	8.56	0.10	37.55	34.81	34.83	0.73
Other viruses	0.61	0.64	0.64	0.03	3.74	3.52	3.70	0.05
Bacteria	0.21	0.33	0.22	0.60	1.19	1.07	1.14	0.12
Archaea	31.30	11.11	31.29	127.78	0.00	0.00	0.00	13.26

Note: Total and percentage sequence read counts for the four highest level taxa assigned by the Epi2ME Fastq WIMP Workflow in undepleted and depleted (bead beating and nuclease treatment prior to nucleic acid extraction) libraries generated from three aliquots of a BoHV1_1-infected bFLC in vitro culture and a nasal swab from a calf experimentally challenged with BoHV-1. PCR-free tagmented libraries were generated with the ONT Field Sequencing Kit and sequenced on a MinION using the rapid base calling option in offline MinKnow sequencing software installed on the MinIT compute module. FASTQ files were subjected to Epi2ME Fastq WIMP analysis.

**Table 3 viruses-14-01859-t003:** Comparison of PCR polymerase bias on BoHV-1 detection sensitivity in MinION sequencing libraries.

Taxonomic Assignment	Sequence Read Counts								
	10 PCR Cycles			20 PCR Cycles			30 PCR Cycles		
BoHV-1 (LongAmp)	81	115	65	59	116	8	206	342	421
BoHV-1 (Q5)	231	36	92	3299	1640	15,179	69,738	65,305	28,423
Bacteria (LongAmp)	21	18	7	612	1273	131	414,892	1,094,002	1,161,715
Bacteria (Q5)	0	0	0	31	19	119	620	505	370
Eukaryote (LongAmp)	7	4	0	183	516	45	1361	2710	3451
Eukaryote (Q5)	2	1	3	43	43	254	2052	1119	877
Other viruses (LongAmp)	0	0	0	1	1	0	163	211	282
Other viruses (Q5)	0	0	0	9	2	15	56	54	25

Note: Libraries were generated from the same nucleic extract (bead beating + nuclease) from BoHV-1-infected bFLCs using the ONT rapid PCR barcoding kit. Either NEB Ultra II Q5 DNA polymerase or NEB LongAmp *Taq* polymerase was used for the PCR step. For each polymerase, libraries were generated using either 10, 20 or 30 PCR cycles. Three libraries were generated for each of these 3 PCR cycle numbers. Read counts for taxonomic assignments following analysis with the Epi2ME Fastq WIMP Workflow of FASTQ files are shown.

**Table 4 viruses-14-01859-t004:** Numbers and percentages of sequence reads assigned to BoHV-1 after MinION sequencing of nasal swabs from experimentally challenged calves.

Calf No.	Day (d) Relative to Challenge								
	d −1	d 0	d 1	d 2	d 3	d 4	d 5	d 6	-ve
	Number of Reads Assigned to BoHV-1								
PBS_1	0	0	0	0	0	0	0	0	0
PBS_2	0	0	0	0	0	0	0	0	0
PBS_3	0	0	0	0	0	0	0	0	0
PBS_4	0	0	0	0	0	0	0	0	0
PBS_5	0	0	0	0	0	0	0	0	0
PBS_6	0	0	0	0	0	0	0	0	0
BoHV1_1	0	1	635	44,864	1779	314	356	1186	1
BoHV1_2	1	2	69,864	3024	204,042	26,519	7	884	2
BoHV1_3	0	0	14	1607	1	220	103	2218	0
BoHV1_4	0	0	174	46,381	478	259	284	47	0
BoHV1_5	0	0	134	1405	385	939	86	110	0
BoHV1_6	0	0	7	554	751	163	355	2300	0
	Percentage of classified reads assigned to BoHV-1								
PBS_1	0.00	0.00	0.00	0.00	0.00	0.00	0.00	0.00	0.00
PBS_2	0.00	0.00	0.00	0.00	0.00	0.00	0.00	0.00	0.00
PBS_3	0.00	0.00	0.00	0.00	0.00	0.00	0.00	0.00	0.00
PBS_4	0.00	0.00	0.00	0.00	0.00	0.00	0.00	0.00	0.00
PBS_5	0.00	0.00	0.00	0.00	0.00	0.00	0.00	0.00	0.00
PBS_6	0.00	0.00	0.00	0.00	0.00	0.00	0.00	0.00	0.00
BoHV1_1	0.00	0.01	45.50	93.04	18.50	32.85	36.00	18.13	1.22
BoHV1_2	0.65	0.55	48.13	4.54	72.73	16.08	1.13	0.41	0.36
BoHV1_3	0.00	0.00	0.01	3.25	0.39	0.60	0.33	0.77	0.00
BoHV1_4	0.00	0.00	1.31	35.90	1.35	0.08	0.11	0.02	0.00
BoHV1_5	0.00	0.00	0.09	2.57	4.74	3.42	1.43	0.71	0.00
BoHV1_6	0.00	0.00	0.00	4.70	1.40	1.83	2.66	0.50	0.00
	Percentage of classified and unclassified reads assigned to BoHV-1								
PBS_1	0.00	0.00	0.00	0.00	0.00	0.00	0.00	0.00	0.00
PBS_2	0.00	0.00	0.00	0.00	0.00	0.00	0.00	0.00	0.00
PBS_3	0.00	0.00	0.00	0.00	0.00	0.00	0.00	0.00	0.00
PBS_4	0.00	0.00	0.00	0.00	0.00	0.00	0.00	0.00	0.00
PBS_5	0.00	0.00	0.00	0.00	0.00	0.00	0.00	0.00	0.00
PBS_6	0.00	0.00	0.00	0.00	0.00	0.00	0.00	0.00	0.00
BoHV1_1	0.00	0.00	8.23	71.40	1.26	1.94	0.97	6.10	0.13
BoHV1_2	0.65	0.81	34.13	1.99	65.94	11.95	1.49	0.47	0.00
BoHV1_3	0.00	0.00	0.01	2.54	0.24	0.37	0.17	0.41	0.00
BoHV1_4	0.00	0.00	0.89	34.64	0.95	0.05	0.08	0.02	0.00
BoHV1_5	0.00	0.00	0.07	2.27	2.68	2.74	0.83	0.38	0.00
BoHV1_6	0.00	0.00	0.00	2.64	1.03	3.35	1.66	0.42	0.00

Note: Epi2ME Fastq WIMP Workflow analysis of rapid basecalled FASTQ files generated using MinION sequencing of nucleic acid extracted from nasal swabs collected from 6 calves challenged with PBS and 6 calves challenged with BoHV-1 on the day prior to the challenge (d-1), the day of the challenge (d 0) and up to d 6 post-challenge. A single flowcell was used for each calf with 12 barcoded libraries run on each flowcell. Where two differently barcoded libraries were run for the same swab (see Appendix A for details), the average of the two libraries is shown here. -ve represents the extraction that was performed on a clean swab, which was included as a negative extraction control in each batch of swab extractions.

**Table 5 viruses-14-01859-t005:** Alignment of the BoHV-1 genome of FASTQ files with from swabs from calf BoHV1_1.

Day Relative to Challenge	Barcode	Number of Aligned Bases (kb)	Number of Unaligned Bases (kb)	Number of Aligned Reads	Number of Unaligned Reads	Bases That Aligned (%)	Average Length of Aligned Reads	Average Length of Unaligned Reads	Average Identity of BoHV-1-Aligned Reads (%)
−1	1	0	3700	0	62,15	0.0	0.0	595.3	0
0	2	0	28,000	0	28,657	0.0	0.0	977.1	0
0	3	0	73,400	0	39,985	0.0	0.0	1835.7	0
1	4	1200	8200	948	8832	12.8	1265.8	928.4	95.5
1	5	546	5253.8	549	5320	9.4	994.9	987.6	95.5
2	6	12,400	6100	14,567	4617	67.0	851.2	1321.2	95.4
2	7	223,600	37,000	83,070	17,099	85.8	2691.7	2163.9	95.4
3	8	3000	163,900	2032	141,366	1.8	1476.4	1159.4	95.4
4	9	624	14,875.1	361	16,133	4.0	1731.0	922.0	95.5
5	10	173	34,026.7	438	36,820	0.5	395.7	924.1	95.5
6	11	1400	22,000	1397	18,351	6.0	1002.1	1198.8	95.5
PBS	12	3.1	390.3	1	773	0.8	3100.0	504.9	94.3

Note: Rapid basecalled FASTQ files from libraries run on a single FLO-MIN106D (R9.4.1) flowcell were aligned to the BoHV-1 genome using ‘Epi2ME Fastq Custom Alignment Workflow’.

## Data Availability

All FASTQ files produced in this study were deposited to the NCBI SRA repository and are available through accession number PRJNA783212. FASTA files of contigs from the BoHV-1 assembly from swabs were deposited to NCBI GenBank and are available under the following accession numbers: OM860299, OM860300, OM860301, OM860302 and OM860303.

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
