# Peer review of "Assessment of Rapid MinION Nanopore DNA Virus Meta-Genomics Using Calves Experimentally Infected with Bovine Herpes Virus-1"

_viruses, 2022, doi:10.3390/v14091859_

Round 1

Reviewer 1 Report

In this paper Esnault et al., reported a novel method to detect the pathogens of BRD. Indeed it is very important to resolve the essence of the pathogens related to BRD with lower cost and at higher efficiency. Though detection of pathogens with metagenomic methods of Nanopore of Technologies has been reported, the application in BRD has not been reported.

However, it would be nice to see a data for validation with different methods, such as PCR. Please verify two or three pathogens with golden method(s).

Author Response

Comments and Suggestions for Authors

In this paper Esnault et al., reported a novel method to detect the pathogens of BRD. Indeed it is very important to resolve the essence of the pathogens related to BRD with lower cost and at higher efficiency.

Reviewer 1. Though detection of pathogens with metagenomic methods of Nanopore of Technologies has been reported, the application in BRD has not been reported.

Authors response: The detection of pathogens in BRD cases with metagenomic methods with Nanopore of Technologies has been reported by Zhang reference [9]. This was cited in the manuscript. ‘This has prompted the increased application of viral metagenomic approaches based on next generation sequencing (NGS) (e.g. Illumina platform) and third generation sequencing (TGS) (e.g. Oxford Nanopore Technologies platform) to BRD-associated virus diagnostics [8,9,19].’ We have now put this Zhang reference immediately after ‘(Oxford Nanopore Technologies platform)’ to make this clearer. We have also discussed details of another Zhang et al. nanopore BRD reference in the discussion now to highlight how our method improves on theirs.

Reviewer 1. However, it would be nice to see a data for validation with different methods, such as PCR. Please verify two or three pathogens with golden method(s).

Authors response: We have now included the BoHV-1 qPCR analysis we conducted on undepleted and depleted nasal swab eluates from the same nasal swabs that were used for sequencing that were collected daily from one of the animals during the 8 day trial. The current gold standard would be considered as qPCR on nucleic acid extracted from undepleted swab eluates. We have included analysis of the relationship between qPCR on undepleted swab eluates and the sequence from the depleted swab eluates. We have published a meetings abstract of another study we conducted where we applied our nanopore viral metagenomics method, with the inclusion of a ds cDNA synthesis step, to nasal swabs taken from cattle that developed BRD naturally in our facility. This abstract is now included in the references. In this abstract (which has been now been submitted as a full paper to another journal) we report detection, using this nanopore viral metagenomics sequencing method, of high prevalence of bovine coronavirus and bovine rhinitis A virus sequences in depleted eluate aliquots from these nasal swabs. We also report that we then confirmed this by qPCR analysis of these two viruses on undepleted eluate aliquots from the same nasal swabs. qPCR positive and negative samples correlated well with sequence positive and negative samples. In this other study, we applied qPCR to undepleted nasal swab eluate in order to more accurately quantify bovine coronavirus and bovine rhinitis A virus.

We did not include these qPCR results in the current paper as we want to publish them as a stand-alone paper.

The bead-beating-DNase/RNase depletion of non-viral nucleic acid hugely increases the ratio of viral to non-viral nucleic acid and, as we and others have shown, is necessary to achieve generation of viral sequence from nasal swabs. But this depletion also lowers the overall amount of virus in the eluate (as we demonstrate in this paper with our experiments on BoHV1 cell culture). So depleted swab eluate used for viral sequencing cannot be used for absolute quantification of viral nucleic acid. qPCR on undepleted swab eluate is probably still the best method to quantify the amount of viral nucleic acid in the swabs. But we have now included in the discussion how addition of encapsidated control virus spike-ins to nasal swab eluates immediately before depletion treatments, may allow estimation of amount of loss of virus due to depletion and thus allow estimation of pre-depletion viral quantities.

Reviewer 2 Report

In this manuscript, the authors assessed whole genome sequencing (WGS) viral metagenomics on the portable MinION sequencer and Epi2ME cloud-based software for same-day, sample to result, identification of a BRD-associated DNA virus (BoHV-1) in nasal swabs from cattle. In shotgun metagenomics, whole DNA is extracted from a sample and exposed to random fragmentation before next-generation sequencing (NGS)

The authors isolated BoHV-1 genomic DNA from nasal swabs of animals challenged with BoHV-1. Initially, they used the Rapid PCR Barcoding commercial Kit (Oxford Nanopore Technologies, Oxford, UK), using the standard Oxford Nanopore protocols to generate the barcoded DNA fragments from the isolated BoHV-1 DNA. They determined that LongAmp Taq, used in the Oxford kit, did not amplify the BoHV-1 genome-specific fragments efficiently and preferentially amplified the non-viral DNA. They then tried the PCR barcoding protocol with either NEB LongAmp Taq or NEB Next Ultra II Q5 polymerase using a slightly modified DNA extraction protocol. Specifically, the cellular and non-viral DNA and RNA were eliminated first by DNAse and RNAse treatments before the protease treatment of nucleocapsid to release the viral DNA. They determined that NEB LongAmp Taq also did not improve the results; however, the NEB Next Ultra II Q6 and a shorter extension time, 30 seconds,  in the PCR parameters, improved the sequence length and the quality significantly.

Comments

This is a complex manuscript but well written. The authors have done an excellent job of explaining how they managed to troubleshoot the problems associated with using LongAmp Taq polymerases for point-of-care/same-day/sample-to-result metagenomic sequence diagnostics

Author Response

Comments and Suggestions for Authors

In this manuscript, the authors assessed whole genome sequencing (WGS) viral metagenomics on the portable MinION sequencer and Epi2ME cloud-based software for same-day, sample to result, identification of a BRD-associated DNA virus (BoHV-1) in nasal swabs from cattle. In shotgun metagenomics, whole DNA is extracted from a sample and exposed to random fragmentation before next-generation sequencing (NGS).

The authors isolated BoHV-1 genomic DNA from nasal swabs of animals challenged with BoHV-1. Initially, they used the Rapid PCR Barcoding commercial Kit (Oxford Nanopore Technologies, Oxford, UK), using the standard Oxford Nanopore protocols to generate the barcoded DNA fragments from the isolated BoHV-1 DNA. They determined that LongAmp Taq, used in the Oxford kit, did not amplify the BoHV-1 genome-specific fragments efficiently and preferentially amplified the non-viral DNA. They then tried the PCR barcoding protocol with either NEB LongAmp Taq or NEB Next Ultra II Q5 polymerase using a slightly modified DNA extraction protocol. Specifically, the cellular and non-viral DNA and RNA were eliminated first by DNAse and RNAse treatments before the protease treatment of nucleocapsid to release the viral DNA. They determined that NEB LongAmp Taq also did not improve the results; however, the NEB Next Ultra II Q6 and a shorter extension time, 30 seconds,  in the PCR parameters, improved the sequence length and the quality significantly.

Reviewer 2 Comments

This is a complex manuscript but well written. The authors have done an excellent job of explaining how they managed to troubleshoot the problems associated with using LongAmp Taq polymerases for point-of-care/same-day/sample-to-result metagenomic sequence diagnostics

Authors response: Thankyou for your supportive and encouraging comments.

Reviewer 3 Report

The authors present an alternative and innovative methodology for diagnosing bovine respiratory disease using metagenomics. Some comments are made to improve the manuscript, qualify the text and make the article more attractive to the reader.

Abstract The abstract is quite confusing. Authors should follow the rule of presenting a brief introduction, objectives, material, and methods, results, and principal conclusions. The way it is written, it is not possible, or at least very confusing, to identify these parts of the abstract. A well-structured and clear abstract is the starting point for the reader to become interested in the article. 

Introduction The introduction is too long. The authors should reduce the introduction and present the article's main points: bovine respiratory complex agents, the importance of rapid diagnosis, diagnostic methods, and objective. I suggest that authors be straight and concise in their presentation. In some paragraphs, the reader feels that they are reading the discussion. 

M&M The authors should make it more straightforward and explain some aspects of M&M. 2.1. The description of the isolate should be reduced. There is no need to describe the signs of the animal where the virus was isolated. Just state that the virus is a virulent field sample isolated from a typical clinical case of IBR with pulmonary complications. The concentration of virus used to inoculate the cells is a bit confusing. It is recommended that the authors turn to MOI (multiplicity of infection). Correct the description of the viral titer. Were the cultures observed at 4x or 40x

2.2. How did the authors confirm that the serology was derived from MDA and not infection? How could this interfere with the experiment?

Results: The results are presented in great detail by the authors. The current format is confusing and does not follow a logical sequence. I suggest that the authors present the animals' clinical results and the methodology developed after that for better understanding. 

3.3 Due to the high cost of the FLO-MIN106D R9.4.1 flow cells at the time of writing, we wanted to develop a protocol that would allow an optimal number of samples to be processed in a single FLO-MIN106D R9.4.1 flow cell. The authors need not mention that they got the idea while writing the article. 

Discussion, The authors use five references in the discussion. The reader feels that the discussion is more a commentary on the results than a proper discussion. This item is essential and should be revised. 

Conclusions The conclusions are not consistent with the results found. The authors mention conclusions unrelated to the paper's primary focus. This should be corrected. 

Author Response

Comments and Suggestions for Authors

The authors present an alternative and innovative methodology for diagnosing bovine respiratory disease using metagenomics. Some comments are made to improve the manuscript, qualify the text and make the article more attractive to the reader.

Abstract The abstract is quite confusing. Authors should follow the rule of presenting a brief introduction, objectives, material, and methods, results, and principal conclusions. The way it is written, it is not possible, or at least very confusing, to identify these parts of the abstract. A well-structured and clear abstract is the starting point for the reader to become interested in the article.

Author’s response: The abstract  has been changed now.

Introduction The introduction is too long. The authors should reduce the introduction and present the article's main points: bovine respiratory complex agents, the importance of rapid diagnosis, diagnostic methods, and objective. I suggest that authors be straight and concise in their presentation. In some paragraphs, the reader feels that they are reading the discussion.

Author’s response: We have tried to simplify and shorten the introduction without compromising the explanation of the work described in the paper and the relatively new and complex but rapidly expanding field of research that our work fits into. We have moved parts of the introduction, which were probably more appropriate for the discussion section, to the discussion.

M&M The authors should make it more straightforward and explain some aspects of M&M. 2.1. The description of the isolate should be reduced. There is no need to describe the signs of the animal where the virus was isolated. Just state that the virus is a virulent field sample isolated from a typical clinical case of IBR with pulmonary complications. Author’s response: This has been done.

The concentration of virus used to inoculate the cells is a bit confusing. It is recommended that the authors turn to MOI (multiplicity of infection). Correct the description of the viral titer.

Author’s response: This has been corrected.

Were the cultures observed at 4x or 40x 

Author’s response: It was 4x magnification.

2.2. How did the authors confirm that the serology was derived from MDA and not infection? How could this interfere with the experiment? Author’s response: This has been changed from MDA (maternally derived antibodies) to just BoHV-1 antibodies.

Results: The results are presented in great detail by the authors. The current format is confusing and does not follow a logical sequence. I suggest that the authors present the animals' clinical results and the methodology developed after that for better understanding. Author’s response: This has been done.

3.3 Due to the high cost of the FLO-MIN106D R9.4.1 flow cells at the time of writing, we wanted to develop a protocol that would allow an optimal number of samples to be processed in a single FLO-MIN106D R9.4.1 flow cell. The authors need not mention that they got the idea while writing the article.

Author’s response: The phrase ’at the time of writing’ has now been deleted.

Discussion, The authors use five references in the discussion. The reader feels that the discussion is more a commentary on the results than a proper discussion. This item is essential and should be revised.

Author’s response: the discussion has been expanded and now includes more references.

Conclusions The conclusions are not consistent with the results found. The authors mention conclusions unrelated to the paper's primary focus. This should be corrected.

Author’s response: The conclusions now hopefully better reflect the paper’s primary focus in they specifically state observations that related to the objective stated in the final paragraph of the introduction.